# Epigallocatechin Gallate for the Treatment of Benign and Malignant Gynecological Diseases—Focus on Epigenetic Mechanisms

**DOI:** 10.3390/nu16040559

**Published:** 2024-02-17

**Authors:** Marta Włodarczyk, Michał Ciebiera, Grażyna Nowicka, Tomasz Łoziński, Mohamed Ali, Ayman Al-Hendy

**Affiliations:** 1Department of Biochemistry and Pharmacogenomics, Faculty of Pharmacy, Medical University of Warsaw, Banacha 1B, 02-097 Warsaw, Poland; gnowicka@wum.edu.pl; 2Centre for Preclinical Research, Medical University of Warsaw, Banacha 1B, 02-097 Warsaw, Poland; 3Second Department of Obstetrics and Gynecology, Centre of Postgraduate Medical Education, 00-189 Warsaw, Poland; mciebiera@cmkp.edu.pl; 4Warsaw Institute of Women’s Health, 00-189 Warsaw, Poland; 5Development and Research Center of Non-Invasive Therapies, Pro-Familia Hospital, 35-302 Rzeszów, Poland; 6Department of Obstetrics and Gynecology, Pro-Familia Hospital, 35-302 Rzeszow, Poland; tomasz.lozinski@pro-familia.pl; 7Department of Gynecology and Obstetrics, Institute of Medical Sciences, College of Medical Sciences, University of Rzeszow, 35-310 Rzeszow, Poland; 8Department of Obstetrics and Gynecology, University of Chicago, 5841 S. Maryland Ave., Chicago, IL 60637, USA; mohamed.ali@bsd.uchicago.edu (M.A.); aalhendy@bsd.uchicago.edu (A.A.-H.)

**Keywords:** EGCG, epigallocatechin gallate, DNA methylation, epigenetic regulation, uterine fibroids, endometriosis, polycystic ovary syndrome, ovarian cancer, endometrial cancer

## Abstract

The most common malignant gynecologic diseases are cervical, uterine, ovarian, vaginal, and vulvar cancer. Among them, ovarian cancer causes more deaths than any other cancer of the female reproductive system. A great number of women suffer from endometriosis, uterine fibroids (UFs), adenomyosis, dysmenorrhea, and polycystic ovary syndrome (PCOS), which are widespread benign health problems causing troublesome and painful symptoms and significantly impairing the quality of life of affected women, and they are some of the main causes of infertility. In addition to the available surgical and pharmacological options, the effects of supporting standard treatment with naturally occurring compounds, mainly polyphenols, are being studied. Catechins are responsible for the majority of potential health benefits attributed to green tea consumption. Epigallocatechin gallate (EGCG) is considered a non-toxic, natural compound with potential anticancer properties. Antioxidant action is its most common function, but attention is also drawn to its participation in cell division inhibition, apoptosis stimulation and epigenetic regulation. In this narrative review, we describe the role of EGCG consumption in preventing the development of benign reproductive disorders such as UF, endometriosis, and PCOS, as well as malignant gynecologic conditions. We discuss possible epigenetic mechanisms that may be related to the action of EGCG.

## 1. Introduction

Among malignant gynecological diseases, cervical and ovarian cancers remain among the main causes of death in women of reproductive age. In turn, less dangerous, mild abnormalities such as uterine fibroids (UFs), endometriosis, or polycystic ovary syndrome (PCOS) can significantly impact women’s quality of life and fertility, with further socio-economic consequences [1]. Notably, there is no effective therapy so far for these benign diseases that would relieve disease symptoms while being fertility-friendly. Moreover, the available drugs often cause side effects and have a risk of disease recurrence after discontinuation of treatment. In the case of malignancies, there are effective therapies, but they cause side effects on the gastrointestinal, hepatic, and hematology systems. The results of observational and epidemiological studies highlight the important role that nutritional factors can play in the treatment and prevention of gynecological diseases, such as endometriosis, UFs, and PCOS, as well as cancers of the reproductive tract. In addition, attention to following a healthy lifestyle and utilizing appropriate micro- and macro-elements consumed from the diet has been increasingly encouraged, considering their involvement in maintaining proper cell and tissue functions and thus preventing many chronic diseases [1,2]. Importantly, fruits, vegetables, and other plant products are a rich source of polyphenols, with flavonoids accounting for a large proportion of the compounds. Almost all categories of flavonoids have anticancer effects due either to their antioxidant or anti-inflammatory properties and probably to their ability to modify the DNA methylation epigenetic profile. However, the epigenetic-mediated effects of this class of polyphenolic compounds on cancer are still not clearly confirmed and remain to be elucidated [3].

### 1.1. The Role of Epigenetics in the Development of Cancers

There is growing evidence that epigenetic alterations such as aberrant DNA methylation or histone modification can play an important role in the pathogenesis of gynecologic diseases and may contribute to adverse cancer phenotypes, including increased cell proliferation, metastasis, chemo-resistance, and immune tolerance [4]. Some studies on gynecological cancers pointed to DNA methylation as a modulator of key genes involved in the regulation of cellular processes. Briefly, the dysregulation of the methylation pattern of genes involved in hormonal responses, cell cycle progression, DNA damage and repair, signal transduction and tumor proliferation, chemo-resistance, and immune tolerance can provide an environment favorable to the maintenance of cancer cells [5]. Other epigenetic mechanisms involving chromatin remodeling, histone modifications, and the modulation of gene expression by non-coding RNAs such as microRNAs (miRs) can also play an important role in the stability of tissue-specific gene expression patterns [6,7]. Histone acetylation, which results from the balance between histone deacetylases (HDACs) and histone acetyltransferases (HATs), is strongly involved in the epigenetic regulation of gene expression. Histone deacetylases specifically play a role in maintaining proper chromatin structure and thus affect heterochromatin silencing, gene transcription, and DNA repair [8,9]. The deregulation of the gene transcription process may be the result of modifications to specific histone sites involved in the organization of chromatin structure. Therefore, understanding the context of epigenetic changes is important both for exploring the pathogenesis of cancer and for developing effective methods for its therapy and prevention.

DNA methylation is one of the most important epigenetic regulatory processes in carcinogenesis, as it significantly contributes to the regulation of both gene expression and phenotypic changes. The main enzymes involved in methylation reactions in mammals are DNA methyltransferases (DNMTs). They catalyze the transfer of methyl groups from S-adenosylmethionine to the 5′ carbon of cytosine bases at sites rich in CpG, in the area of gene promoters, contributing to changes in gene expression. DNMT3A and DNMT3B are associated with de novo DNA methylation, and DNMT1 is responsible for DNA methylation inheritance [10]. DNA methylation aberration has been shown to play a significant role in malignant gynecological cancer. For instance, hypermethylation was associated with the inactivation of several pathways involved in ovarian cancer development, including DNA repair, cell cycle regulation, apoptosis, cell adherence, and detoxification pathways [11]. Similarly, in cervical cancer, the hypermethylation of gene promoters involved in cell adhesion, apoptosis, or the p53 signaling pathway has been recognized [12].

As epigenetic modifications are reversible, an increasing number of therapeutic agents targeting epigenetic alterations are being developed and represent an exciting area of research, including DNMT inhibitors and HDAC inhibitors [13]. Several DNMT inhibitors have been approved by the U.S. Food and Drug Administration and are used to treat a variety of blood and solid malignancies [14]. Both HDAC inhibitors and DNMT inhibitors have been studied in ovarian and cervical cancers as single agents or in combination with other therapies; however, it is worth mentioning that not all the results from in vitro studies can always be confirmed or extended to the clinical trial phase [13,15].

### 1.2. New Therapeutic Options for Gynecological Diseases—The Role of Dietary Compounds

Recently published reviews indicate a growing interest in dietary modifications as a support for or element of conventional treatment methods [16,17,18]. Most supporting evidence regarding the beneficial effects of dietary ingredients comes from epidemiological reports. Therefore, results from experimental studies are needed to confirm, at the molecular level, the independent impact of individual nutrients on the development of gynecological diseases. Therapeutic strategies used to prevent and reduce the risk of certain gynecological disorders are mainly limited to hormonal treatment; therefore, examining the relationship between these disorders and diet will introduce women to new therapeutic perspectives. Controlling for various confounding factors may prove important to precisely determine the etiological role of diet.

There is an increasing interest in the use of dietary compounds as a support for the treatment of mild gynecological diseases [19,20]. Biologically active ingredients in medicinal plants may be a source of new drug candidates which slow down the progression of, among others, fibroids and endometriosis [21,22]. In malignant gynecological cancers, the interaction of dietary factors and carcinogenesis is still controversial because most of the data are conflicting and limited. However, the results of a recently published cross-sectional study based on a dietary questionnaire showed potential negative correlations between micronutrient intake and the development of gynecological cancers, i.e., between vitamin B1 and cervical cancer; zinc and ovarian cancer; and potassium and endometrial cancer. Furthermore, a positive association was observed between calcium intake and cervical cancer and endometrial cancer and between sodium intake and endometrial cancer [23]. Among others, great attention is paid to the role of natural antioxidants, such as carotenoids, present in vegetables and fruits, indicating their beneficial effects on the course of gynecological cancers [24,25]. Although we expect to gain beneficial outcomes from increased consumption of fruits and vegetables, not all studies confirm this effect in the context of gynecological cancers or benign diseases. A case–control study conducted in China found that the consumption of fresh vegetables and green tea provided protection against cervical cancer, while no such association was reported with the consumption of fruits, eggs/milk/meat, or soy [26]. This is in contrast to a European study, where the results indicated the protective effect of fruit consumption but not vegetable consumption [27]. The mechanism of vegetables’ protective effects against endometrial cancer is probably based on changes in estrogen metabolism, the induction of antioxidant mechanisms, and the activation of the immune system, while phytoestrogens contained in the Mediterranean diet act by competing with endogenous estrogens, antagonizing their effect on the endometrium. In turn, animal products induce pro-inflammatory markers such as CRP, which may be associated with the development of endometrial cancer [28,29,30]. Studies of the female population in Italy have shown that the risk of developing endometriosis is inversely proportional to the consumption of vegetables and fruits [31]. Similarly, the beneficial effect of enriching the diet with sources of carotenoids, polyphenols, and quercetin was associated with a reduction in symptoms and an improvement in the quality of life in patients with UFs [21].

Some studies focused on single compounds’ effects and also provided promising data. Patients with pelvic pain and endometriosis treated with vitamins C or E reported less pain, dysmenorrhea, and dyspareunia compared to the placebo group. Additionally, a significant reduction in inflammatory markers was found compared to patients who did not take vitamins [32]. Therefore, the effects of alleviating the clinical symptoms of endometriosis may be the result of the antioxidant properties of vitamins. In randomized trials, vitamin D supplementation significantly reduced pelvic pain in women with endometriosis and was related to reduced growth of UFs [33,34,35]. This was not confirmed in another study, where a significant reduction in dysmenorrhea and/or pelvic pain as a result of vitamin D administration in endometriosis patients has not been observed [36].

### 1.3. Methods

This article reviews the available literature discussing the role of EGCG, a component of green tea, in the most common benign and malignant gynecological diseases. The electronic databases used were PubMed, Google Scholar, and Clinictrial.gov. The literature was searched for relevant clinical studies and observational studies, as well as data from in vivo and in vitro studies. Full-text papers in English published from January 2003 to November 2023 were considered. The search terms included “green tea”, “EGCG”, “Epigallocatechin-3-gallate”, “EGCG mechanisms”, “pharmacokinetics”, “pharmacodynamics”, “bioavailability”, “metabolism”, “endometriosis”, “uterine leiomyoma”, “uterine fibroid”, “polycystic “ovary syndrome”, “PCOS”, “polycystic ovary syndrome”, “ovarian cancer”, “cervical cancer”, and “endometrial cancer”. A summary of papers on the possible impact of EGCG administration (alone or in combination with other compounds) on the prevention of gynecological diseases is presented in Table 1 and Table 2. The characteristics of registered clinical trials with EGCG treatment are shown in Table 3. Where data were available, we also described the possible influence of epigenetic mechanisms.

## 2. Epigallocatechin Gallate as an Active Phenolic Compound of Green Tea

Therapeutic strategies used to prevent and reduce the risk of certain gynecological disorders are mainly limited to hormonal treatment; therefore, examining the relationship between these disorders and diet will introduce women to new therapeutic perspectives. Controlling for various confounding factors may prove important to precisely determine the etiological role of diet. Fruits, vegetables, and other plant products used in beverages are a rich source of polyphenols, with flavonoids accounting for a large proportion of the polyphenolic compounds [3]. Due to their ability to modify the DNA methylation profile, flavonoids should be recognized for their use in cancer prevention. However, until now, there have been practically noevidence on the direct effect of particular polyphenols on gene methylation in gynecologic cancers such as cervical or ovarian cancer or benign diseases such as UFs or endometriosis [3].

Tea from the Camellia sinensis plant is one of the most commonly consumed beverages in the world. It has been used in traditional Chinese medicine since ancient times as a remedy for many ailments. Its health-promoting properties are constantly scientifically confirmed. Among dietary ingredients, green tea has a high catechin content, and one of the major components of catechins in green tea is epigallocatechin-3-gallate (EGCG), an ester of epigallocatechin and gallic acid [37]. Other important green tea polyphenols are epicatechin-3-gallate, epigallocatechin, gallocatechin, and epicatechin. Their main function is to scavenge free radicals, based on their high antioxidant properties [38]. Catechins also possess metal-chelating properties, and they may also have pro-oxidative and cytotoxic effects. Catechins can interact directly with proteins and phospholipids and thereby regulate signal transduction pathways. Their high affinity to lipid bilayers has been demonstrated, which may explain their penetration into cancer cells [39]. In vitro and in vivo studies have revealed the anticancer properties of tea polyphenols. However, it is mainly EGCG that is responsible for the majority of health benefits attributed to green tea [40,41,42]. Due to their ability to modify the DNA methylation profile, flavonoids, such as EGCG, are considered for use in cancer prevention. However, until now, data on the direct effect of EGCG on gene methylation in gynecologic cancers such as cervical or ovarian cancer have been scarce. Studies on cell lines, unfortunately not covering the discussed gynecological diseases, have shown that EGCG acts mainly as a hypomethylator, inhibiting the DNMT enzyme by creating hydrogen bonds in this protein, which results in the reactivation of genes silenced by methylation [43]. Also, the results of molecular docking studies suggest that EGCG may also competitively inhibit other epigenetic enzymes such as HDAC2, HDAC3, HDAC4, HDAC7, and EZH2 (Figure 1) [44]. In-depth research and evaluations of data from large-scale analyses are needed to show that such a direct or indirect epigenetic effect of EGCG exists in the case of the discussed gynecological diseases.

### EGCG Bioavailability

The main issue discussed in the context of the use of EGCG in therapy is its metabolism and bioavailability [45]. A single-dose EGCG pharmacokinetic evaluation was studied. Serum EGCG greater than 1 μM was observed with oral doses greater than one gram of EGCG (dose: 1600 mg; Cmax = 3392 ng/mL; range: 130–3392 ng/mL). In addition, it was found that after 1.3–2.2 h, the maximum concentration was reached [46]. The safety and kinetics of EGCG and decaffeinated green tea extract were also evaluated by Chow et al. [47]. EGCG intake at doses of 400 and 800 mg resulted in serum concentrations of both free and total EGCG in the high nanomolar range. The chronic administration of a dose of 800 mg resulted in an increase in the bioavailability of EGCG with only minor gastrointestinal side effects. The content of polyphenols in one sachet of green tea is about 80–100 mg, which translates into 25–30 mg of EGCG. Therefore, a therapeutic dose may be provided, either as a dietary supplement with a condensed high amount of EGCG or as a pro-EGCG compound with enhanced bioavailability. The presence of a hydroxyl group in EGCG determines its low bioavailability in the human body [48,49]. The alkaline environment causes the rapid oxidation of EGCG and is not conducive to protons on the phenolic groups, resulting in the production of a phenolate anion that acts on electrophilic agents such as free radicals in the body [50]. To prevent the breakdown of EGCG in the body, various methods of delivery are designed, e.g., the encapsulation or formation of a prodrug [51]. Pro-EGCG was shown to act as an angiogenesis inhibitor in in vitro and in vivo models of endometrial cancer [52]. Additionally, pro-EGCG was also found to be more bioavailable and stable than traditional EGCG extract in breast cancer in vitro and in vivo [53]. The antiproliferative, antiangiogenic, and antifibrotic effects of pro-EGCG were confirmed in human leiomyoma cell lines [54]. Clinical trials are needed to understand the mechanism and effectiveness of this new drug in patients.

## 3. EGCG in the Treatment of Benign Gynecological Diseases

### 3.1. Uterine Fibroids

Uterine fibroids (UFs), which are benign neoplasms, occur in women of reproductive age with various frequencies, depending on the population [55,56]. They are the main source of gynecological and reproductive dysfunctions, causing a number of adverse symptoms, from menorrhagia and pelvic pain to infertility, recurrent miscarriages, and preterm birth [57]. In addition to the Afro-American race, the risk factors for UFs include older age, premenopausal state, nulliparity, a family history of UFs, hypertension, and obesity [58]. Environmental factors, including dietary components, also play an important role. Among them, vitamin D deficiency [59,60,61], excess levels of tocopherol [62], and exposure to endocrine-disrupting chemicals [63] might be of interest. The characteristic over-response of UFs to estrogen and progesterone induces cell proliferation, the inhibition of apoptosis, and the formation of the extracellular matrix (ECM). Large amounts of ECM with high levels of collagen and fibronectin secreted by fibroblasts contribute to tumor cell signaling and expansion [64,65]. Other known proteins involved in those processes are growth factors, e.g., fibroblast growth factor 2 (FGF2), vascular endothelial growth factor (VEGF), transforming growth factor β (TGF β), and activin-A, which impact the pro- and anti-angiogenic properties of the altered tissue [66,67]. The therapy for symptomatic UFs includes medical, surgical, and radiological interventions. A hysterectomy is one of the most effective treatment modalities for such lesions. However, alternative solutions are now available that preserve fertility and help avoid invasive surgery [55]. These days, clinicians are even more inclined to choose less invasive types of surgeries, but the transition is still ongoing. The most effective pharmacological treatments include gonadotropin-releasing hormone (GnRH) analogs (both agonists and antagonists) and selective progesterone receptor modulators (SPRMs) [68]. Non-invasive methods of thermal ablation and uterine artery embolization have been gaining popularity recently [69,70].

#### 3.1.1. Epigenetic Mechanisms in Uterine Fibroid Development—An Overview

An aberrant methylation pattern is often associated with silencing the expression of tumor suppressor genes, which may lead to tumor development and progression [71]. Changes in the genome methylation/demethylation profile in uterine fibroid cells were observed [72,73]. For instance, hypomethylation was recognized within the HOXA13 gene, which encodes a class of transcription factors called homeobox genes, resulting in the upregulation of transcription in UFs compared to normal tissue [74]. The epigenetic mechanisms involved in uterine fibroid development are also associated with the activation of important transduction signaling pathways such as Wnt/β-catenin and Wnt/MAPK [75]. The effects of 5-Aza-2′-deoxycytidine (5-aza-Cd), a DNMT inhibitor, have been studied in uterine leiomyoma and adjacent myometrium tissues and human uterine leiomyoma primary (HULP) cells. Carbajo-Garcia et al. recognized a higher expression of DNMT1 in uterine leiomyoma vs. normal myometrium [76]. After 5-aza-Cd treatment, the expression of proliferating cell nuclear antigens (PCNAs) and ECM proteins (such as collagen, PAI-I, fibronectin) was significantly decreased. A lower expression of genes involved in angiogenesis, proliferation, and invasions, namely Wnt1-inducible-signaling pathway protein 1, protooncogene c-MYC, and matrix metalloproteinase 7 (MMP7), the targets of the Wnt/β-catenin pathway, was also observed. The results suggested that gene demethylation achieved via DNMT inhibitor treatment may inhibit uterine fibroid growth. In other studies in human and mouse endometrial cells, the demethylation-related effects of 5-aza-Cd included cell cycle arrest, the inhibition of cell differentiation, and cell death [77,78]. DNA hypomethylation was also shown to be involved in the regulation of estrogen and estrogen-responsive genes in mesenchymal stem cells [63]. A recently published paper described the relationship between progesterone signaling and DNA methylation and its role in leiomyoma growth [79]. The authors observed the hypermethylation of the progesterone receptor gene (PGR) and the inhibition of PGR expression in uterine leiomyoma stem cells (LSCs). 5-Aza treatment stimulated PGR expression in LSCs and was involved in shrinking UFs in a xenograft mouse model. The results also showed that the PGR acted as an upregulator of the methylation enzymes TET1 and TET2 (Ten-eleven translocation methylcytosine dioxygenase 1 and 2). Moreover, PGR deficiency increased DNA methylation levels around gene body regions in leiomyoma cells [80]. Therefore, it seems that PGR activation may contribute to the global loss of DNA methylation during LSC differentiation.

More information on the role of methylation in the development of UFs will undoubtedly be provided by the results of an ongoing observational study (NCT04214457) [81]. This prospective, multicenter biomedical study, including patients with a surgical indication of hysterectomy or laparoscopic or laparotomic myomectomy due to the diagnosis of myometrial tumors (leiomyoma/leiomyosarcoma), is being conducted in Spain. Next-Generation Sequencing (NGS) technology will be applied for the detection of DNA mutations and the recognition of chromosomal instability. The main aim of this trial is to identify specific genetic markers for leiomyomas and leiomyosarcomas in a group of 1000 participants. However, the secondary outcome measures involve the identification of methylation patterns that could identify tumorigenic processes.

Several studies have revealed differences in the miRNA expression of uterine leiomyoma and myometrial tissue. For instance, significant roles in UF pathogenesis were demonstrated for miR-150-5P, miR-21a-5p, the miR-29 family, and miR-let-7 [82,83,84]. Additionally, an upregulation of miR-127-3p, miR-28-3p, and miR-30b-5p was found in UFs [85]. These molecules are involved in cell proliferation, ECM turnover, and angiogenesis, which are processes that are disturbed in UF cells [6]. However, data concerning the regulation of miRNA and its gene targets in UFs are still insufficient. It was suggested that fibroid development could be determined by early and late epigenome–environment interactions, which affect the shaping of the developing epigenome of target genes [86]. At a later stage of development, this silent epigenomic reprogramming may be activated by several factors. Exposure to bisphenol A was shown to induce epigenomic reprogramming in specific genes and chromatin states in the neonatal liver to accelerate the acquisition of the adult epigenetic signature. This reprogramming was transcriptionally silenced until triggered later in life by high fat, fructose, and cholesterol intakes, characteristic of a Western-style diet [86]. It is recognized that lifestyle and nutrition play an important role in both the etiopathogenesis and prevention of cancer by influencing epigenetic events. As these processes are reversible, we can influence their directions. Therefore, it is important to explore areas of new therapies based on epigenetic modifiers.

#### 3.1.2. EGCG Mechanisms of Action in Uterine Fibroids

Detailed studies on the molecular pathway associated with green tea consumption and the occurrence of UFs are lacking. However, the results of in vitro studies and data from animal models indicate several main directions of action of ECGC, including apoptosis, proliferation, or enzymatic activity impacts (Table 1 and Table 2). Proliferation-inhibiting and apoptosis-inducing effects were demonstrated in Eker rat and human leiomyoma cells treated with EGCG [54,87]. Additionally, in human leiomyoma cells, green tea extract modified catechol-O-methyltransferase (COMT) expression and enzyme activity [88]. As COMT catalyzes the S-adenosyl-L-methionine-dependent methyl conjugate of the hydroxyl group of catechol estrogens, the regulation of its activity may impact estrogen action and play a role in UF development. Higher levels of COMT were confirmed in leiomyoma tissues compared to the normal myometrium [89]. The excessive activity of COMT triggers the transformation of 2-hydroxyestradiol (antiestrogen) into 2-methoxyestradiol (proestrogen). The created hyperestrogenic environment may contribute to cell proliferation and increase UF development. In vitro studies showed that the ability of green tea extract to reduce the expression of the COMT enzyme contributed to the inhibition of leiomyoma cell proliferation [88]. However, another in vivo study showed a 24% increase in COMT activity after the ingestion of EGCG [90]. Due to the fact that EGCG is recognized as a substrate of COMT, it was suggested that the regulation of the COMT enzyme might be a key mechanism responsible for the beneficial effect of EGCG on leiomyoma cells [91]. EGCG was found to act through multiple signal transduction pathways to inhibit HuLM cell proliferation and induce apoptosis. Cells treated with EGCG at a concentration of ≥50 μM had significantly decreased expression of PCNA, CDK4, and BCL-2, as well as an increased expression of pro-apoptotic BAX [87,92]. These results suggest that EGCG could be considered an element in UF therapy. Zhang et al. observed significantly decreased PCNA and Cdk4 protein levels at ≥50 μM of EGCG in Eker leiomyoma tumor-3 (ELT3) cell cultures. Moreover, the in vitro results were confirmed in an in vivo model of athymic nude mice, where the PCNA and Cdk4 protein levels were also significantly reduced after 1.25 mg/day of EGCG treatment [87]. A study on quails involving the supplementation of 200 or 400 mg of EGCG/kg in their diet for 12 months demonstrated a significantly decreased number and size of leiomyomas compared to the control group. The EGCG treatment also resulted in significantly lower serum and liver malondialdehyde and TNF-α concentrations, which is of great interest as it might also be involved in UF pathophysiology [93]. It is also worth highlighting that vitamin D and EGCG also present interesting data [94,95].

**Table 1 nutrients-16-00559-t001:** Therapeutic effects of active ingredients in green tea in cell culture studies.

Disease	Intervention	Cell Line	Relevant Findings	Reference, Year
Uterine fibroids	10–200 μM of EGCG,	HuLM cells	Inhibited growth, decreased proliferation, and decreased gene and protein expression of the PCNA	[96], 2023
	1, 10, 50, 100, and 200 µM of EGCG	HuLM cells	Decreased gene expression or protein levels of fibronectin, COL1A1, PAI-1, CTGF, and ACTA2	[97], 2023
	5, 10, and 25 μM of EGCG, Pro-EGCG, and Pro-EGCG analogs	HuLM cells	Antiproliferative, antiangiogenic, and antifibrotic activities	[54], 2016
	100 µM of EGCG	HuLM cells	Antiproliferative activity and decreased PCNA, Cdk4, and COMT protein levels	[88], 2014
	0.1, 1.0, 10, 50, 100, and 200 µM of EGCG	HuLM cells	Inhibited proliferation, induced apoptosis, decreased the expression of PCNA, CDK4, and BCL2, and increased the expression of BAX	[92], 2010
	0, 1.0, 50, 100, and 200 μM	ELT3 cells	Reduced PCNA and Cdk4 protein levels, inhibited proliferation, and induced apoptosis	[87], 2010
Endometriosis	0–300 μM of EGCG	Primary human endometrial stromal cells;	Increased expression of Nmnat1 and Nmnat3, nicotinamide nucleotide adenylyltransferases	[98], 2021
	50 and 100 µM EGCG	Primary endometrial and endometriotic stromal cells	Inhibited proliferation, migration, and invasion	[99], 2014
	40, 80, and 100 µM of EGCG	Primary human endometrial epithelial cells	Reduced proliferation and increased apoptosis	[100], 2013
	10–50 μM of EGCG	Human microvascular endothelial cells	Inhibited angiogenesis and suppressed VEGFC/VEGFR2 expression and signaling pathway	[101], 2011
PCOS	0–10 μM of EGCG	Human granulosa-like tumor cell line and KGN	Upregulated steroidogenic acute regulatory protein (StAR) expression and increased progesterone production	[102], 2023
Endometrial cancer	20–60 μM of EGCG or ProEGCG	AN3 CA and RL95–2	Antiproliferative effect and induced apoptosis	[103], 2020
	20–60 µM of pro-EGCG	AN3CA and RL95-2	Inhibited angiogenesis and decreased VEGFA secretion through inhibiting PI3K/AKT/mTOR/HIF1α signaling pathway	[52], 2018
	100, 125, and 150 μM of EGCG	HEK-293, Ishikawa cells, and primary endometrial adenocarcinoma cells	Inhibited proliferation, downregulated estrogen receptor α, progesterone receptor, proliferating cell nuclear antigen, and cyclin D1, increased apoptosis, upregulated Bax, downregulated Bcl2, and induced ROS and oxidative stress in endometrial cancer cells	[104], 2013
	100 μM of EGCG	Ishikawa cells	Inhibited proliferation and induced apoptosis	[105], 2012
Ovarian cancer	5, 10, 20, 40, and 80 µg/mL of EGCG	SKOV3, CAOV-3, and NIH-OVCAR-3	Inhibited proliferation, induced apoptosis, upregulated Bax and caspase-3, and downregulated Bcl-2	[106], 2020
	20–100 μg/mL of EGCG	SKOV3	Inhibited proliferation, induced apoptosis, and downregulated NF-κB, p65, and IκB-α	[107], 2012
	20–40 μM of EGCG	HEY and OVCA 433	Inhibited proliferation and induced apoptosis	[108], 2006
	25, 50, and 100 µM of EGCG	SKOV-3, OVCAR-3, and PA-1	Induced apoptosis, upregulated P21 and Bax, and downregulated BCL-XL and PCNA	[109], 2004
Cervical cancer	0–100 µM of EGCG	HeLa, SiHa	Suppressed TGF-β-induced EMT and decreased ROS levels	[110], 2021
	50–100 µM of EGCG	HeLa	Decreased global DNA methylation	[111], 2020
	250–500 µM of EGCG	HeLa, C33A, and WI-38	Induced cell cycle arrest and apoptosis and inhibited cell growth	[112], 2019
	0–100 µg/mL of EGCG	HeLa, CaSki, and C33A	Inhibited cell proliferation	[113], 2019
	60 µg/mL of EGCG	HeLa	Cytostatic effect but no cytotoxic effect	[114], 2018
	25–100 of EGCG µg/mL of	HeLa	Inhibited cellular proliferation and induced apoptosis	[115], 2018
	50 µg/mL of EGCG	HeLa	Free radical scavenging properties, increased activity of SOD and GPx, and inhibited proliferation	[116], 2017
	0, 25, and 50 µM of EGCG	HeLa	Inhibited proliferation and induced cell cycle arrest	[117], 2015
	25 µM of EGCG	HeLa	Inhibited DNA methyl-transferases and histone deacetylases	[118], 2015
	25 µM of EGCG	HeLa	Inhibited cell growth, inhibited cell survival, and induced apoptosis through NFκB p65, COX-2, p-Akt, and p-mTOR signaling	[119], 2014
	1, 10, 25, and 50 µM of EGCG	HeLa	Induced antiproliferative action and reduced mRNA expression of FTS	[120], 2013
	1–100 µM of EGCG	HeLa	Induced apoptosis and inhibited invasion and migration	[121], 2012
	10 µM of EGCG	Hela	Downregulation of genes involved in the stimulation of proliferation, adhesion, motility, and invasion processes and reduced adhesion and proliferation rates	[122], 2012
	5, 10 µg/mL of EGCG	PMBC and cervical carcinoma tissue	Increased apoptosis	[123], 2011
	0–50 µg/mL of EGCG	TCL1, HeLa, and Me180	Inhibited cell growth and induced apoptosis	[124], 2010
	25 and 50 µM of EGCG	HeLa and CaSki	Inhibited cell growth and proliferation	[125], 2008
	80 µg/mL and 100 µM of EGCG	HeLa and HepG2	Inhibited cell migration	[126], 2006

HuLM, Human Uterine Leiomyoma Cells; ELT3, Eker Rat Tumor-Derived Uterine Leiomyoma Cell Line; PMBC, Peripheral Blood Mononuclear Cells; PCNA, proliferating cell nuclear antigen; COL1A1, collagen; PAI-1, plasminogen activator inhibitor-1; CTGF, connective tissue growth factor; ACTA2, actin alpha 2, smooth muscle; COMT, catechol-O-methyltransferase; StAR, steroidogenic acute regulatory protein; NF-κB, nuclear factor kappa-light-chain-enhancer of activated B cells; HIF-1α, hypoxia-inducible factor; VEGF, vascular endothelial growth factor; Bcl-2,B-cell lymphoma 2; TGF-β, transforming growth factor beta; EMT, epithelial-to-mesenchymal transition; ROS, reactive oxygen species; SOD, superoxide dismutase; GPx, glutathione peroxidase; FTS, Fused Toes Homolog; COX, Cyclooxygenase.

#### 3.1.3. EGCG Treatment of Uterine Fibroids

The amount of data concerning the use of EGCG in patients with UFs is growing these days (Table 2). In a double-blinded, placebo-controlled randomized clinical trial, 800 mg of green tea extract (45% EGCG) was recognized to be a safe therapeutic agent for symptomatic women with UFs [127]. Porcaro et al. reported 4 months of combined vitamin D and EGCG supplementation in symptomatic women with myomas [128]. As a result, the total fibroid volume and severity of the symptoms of UFs were significantly decreased in the treated group. Numerous studies, also published by our team, have confirmed a correlation between vitamin D deficiency and the development of UFs [59,129,130,131]. However, it was noticed that the combined effect of EGCG and vitamin D was more beneficial than their separate administration. Recently, the results of a pilot, prospective, monocenter study were published. The study involved the simultaneous administration of tea catechin and vitamin D to a group of 16 women with UFs in late reproductive life. The patients received a total dose of 300 mg of EGCG, 10 mg of vitamin B6, and 50 mg/day of vitamin D for 3 months. A significant reduction in tumor size and menstrual flow length was observed [132]. The same dose of supplements was administered to a group of 41 women at childbearing age with at least one uterine fibroid. After 4 months, a significantly decreased volume of UFs was seen in the study group compared to the controls without treatment. Pelvic pain and heavy bleeding were also reduced after EGCG administration [133]. Such beneficial effects were not observed in a non-randomized, observational study by Biro et al., where 25 women were treated for 6 months with capsules containing 780 mg of green tea extract (EGCG, 390 mg; ascorbic acid, 60 mg; piperine, 15 mg; caffeine 3 mg). The results revealed no significant changes in the size or number of fibroids [134]. The above-mentioned studies showed that the therapeutic effect of EGCG on UFs may be more beneficial with the simultaneous administration of vitamin D. Inhibiting cell proliferation and the ability to induce apoptosis are some of the mechanisms of action of both vitamin D and EGCG [54,135,136,137]. The antiproliferative action of EGCG was achieved through the downregulation of PCNA, CDK4, and BCL-2 [54,92]. The downregulation of PCNA and BCL-2 that resulted in lower cell proliferation was also observed after vitamin D supplementation in an animal model of uterine fibroids [138]. COMT is another protein influenced by both EGCG and vitamin D. Vitamin D alone suppresses COMT expression and activity in human uterine leiomyoma (HuLM) cells [139]. Likewise, decreased COMT activity was seen after EGCG treatment in a wild-type HuLM cell culture [88]. The use of vitamin D and EGCG combinations, which act on the same as well as completely different pathways involved in the pathogenesis of UF, may become an interesting approach in the therapeutic management of UFs. According to ClinicalTrials.gov, clinical trials carried out before 4 December 2023 were found (Table 3). Two of the presented clinical trials were conducted in Italy, and two studies were performed in the United States. Three studies are recruiting for testing EGCG alone or in combination with other drugs: vitamin B6, and D-chiroinositol ([140,141] NCT05409872, NCT05448365). The impact on fertility will be assessed with 1650 mg/day of green tea extract (with 45% EGCG) for up to 7 months. Moreover, the volume of UFs will also be measured ([142] NCT05364008). In another study, 60 participants with UFs will use tablets with epigallocatechin gallate, vitamin D, D-chiroinositol, and vitamin B6 for 3 months. Apart from measuring the volume of UFs, the researchers will determine changes in the expression of ki67, ER, and PGR genes and the phosphorylation of the VEGF receptor (NCT05448365). A study on the pharmacokinetics and hepatic safety of EGCG will evaluate several biochemical parameters such as total bilirubin, alanine aminotransferase (ALT), aspartate aminotransferase (AST), alkaline phosphatase, beta-human chorionic gonadotropin (βhCG), follicle-stimulating hormone (FSH), luteinizing hormone (LH), antiMüllerian hormone (AMH), and estrogen in serum ([143] NCT04177693). This trial will also bring new data on the level of catechins (EGCG, EGC, ′-O-methyl-epigallocatechin) after receiving 800 mg of EGCG (as a green tea extract) for up to 2 months, giving us valuable data about the bioavailability of this compound.

**Table 2 nutrients-16-00559-t002:** Characteristics of papers on the possible impact of EGCG administration (alone or in combination with other compounds) in the prevention of gynecological diseases in animal models and human studies.

Disease	Intervention	Total Number of Subjects	Relevant Findings	Reference, Year
Uterine fibroids	800 mg of EGCG alone, with clomiphene citrate, 100 mg, for 5 days, or with letrozole, 5 mg, per day for 5 days	39 women	No adverse side effects and no liver toxicity or folate deficiency	[144], 2023
	EGCG 300 mg, vitamin B6 10 mg, and vitamin D3 50 µg per day for 3 months	16 women	Reduction in fibroid size and decrease in menstrual flow length	[132], 2022
	EGCG 150 mg, vitamin D3 25 µg, and vitamin B6 5 mg per day for 3 months	30 women	Decreased UF symptoms and improved parameters of quality of life	[145], 2022
	Capsule containing EGCG—390 mg, ascorbic acid—60 mg, piperine—15 mg, and caffeine—3 mg per day for 6 months	25 women	No changes in UF size and symptoms and no adverse side effects.	[134], 2021
	25 μg of vitamin D3 + 150 mg of EGCG + 5 mg of vitamin B6 per day for 4 months	95 women	Reduced volume of fibroids and improved parameters of quality of life	[133], 2021
	30 μg of vitamin D3 + 300 mg of EGCG + 10 mg of vitamin B6 per day for 4 months	30 women	Decreased volume of fibroids and improved parameters of quality of life	[128], 2020
	800 mg of green tea extract (45% EGCG) per day for 4 months	33 women	Reduced fibroid volume and symptom severity, decreased blood loss, and improved parameters of quality of life	[127], 2013
	1.25 mg/mouse of EGCG per day for 8 weeks	20 athymic nude mice	Reduced volume and weight of fibroids	[87], 2010
	200 or 400 mg/kg of EGCG per day,= for 12 months	180 Japanese quail	Decreased serum and liver malondialdehyde and TNF-α concentrations and decreased size of UFs	[93], 2008
Endometriosis	50 mg/kg of EGCG or ProEGCG per day for 21 days	20 C57BL/6 mice	Reduced endometriotic lesion sizes and overexpression of NMNAT1 and NMNAT3 after ProEGCG treatment	[98], 2021
	8.333 mg/mL of EGCG, 0.3 μg/μL of decitabine, or both per day for 16 days	36 BALB/c female nude mice	Inhibition of endometrial lesion growth, increased E-cadherin expression, andreduced DNA methylation of the E-cadherin promoter	[146], 2020
	50 mg/kg of EGCG per day for 3 weeks	30 SCID mice	Inhibition of angiogenesis and suppressed VEGFC/VEGFR2 expression and signaling pathway	[101], 2011
	20 or 100 mg/kg of EGCG per day for 4 weeks	56 BALB/c mouse model	Reduced size of endometriotic lesions and vascular density, decreased cell proliferation, and increased apoptosis	[100], 2013
	50 mg/kg of EGCG per day for 21 or 8 days	40 nude mice	Prevention of fibrosis progression in endometriosis	[99], 2014
	50 mg/kg of EGCG per day for 2 weeks	30 SCID mice	Reduction in number and size of endometrial microvessels, inhibition of angiogenesis, and increase in apoptosis	[147], 2009
	50 mg/kg of pro-EGCG or 50 mg/kg of EGCG per day for 4 weeks	32 NOD-SCID mice	Inhibited growth and angiogenesis and increased apoptosis	[148], 2013,
	50 mg/kg/day of EGCG per day for 2 weeks	40 Swiss nude mice	Inhibited proliferation, migration and invasion	[99], 2014
	50 mg/kg of EGCG per day for 3 weeks	36 BALB/cnude mice	Inhibited angiogenesis and suppressed EGFC/VEGFR2 expression	[101], 2011
PCOS	500 mg green tea tablet per day for 3 months	15 women	Reduction in weight, BMI, and waist and hip circumference	[149], 2020
	500 mg green tea leaf powder tablet per day for 45 days	45 women	Reduction in BMI, body weight, waist circumference, and body fat percentage.	[150], 2017
	500 mg green tea tablet per day for 3 months	60 women	Reduction in body weight, serum insulin, and free testosterone	[151], 2017
	1.5 cups of 2% Lung Chen tea (equivalent to 540 mg of EGCG) per day for 3 months	34 obese women	No significant effect on body weight, BMI, body fat, total testosterone, SHBG, free androgen index, DHEA-S, FSH, or LH	[152], 2006
	50, 100, or 200 mg/kg of green tea extract per day for 14 days	40 Wistar rats	Reduced CRP and IL-6 serum levels and improved ovarian function	[153], 2014;
Endometrial cancer	50 mg/kg of EGCG or ProEGCG per day for 5 weeks	15 athymic nude mice	Reduced tumor growth and downregulation of NOD1 and NAIP	[103], 2020;
	65 mg/kg EGCG per day for 2 weeks	7 Syrian golden hamsters	Inhibited VEGF expression	[154], 2008
	50 mg/kg of pro-EGCG per day for 35 days	15 nude mice	Inhibited angiogenesis and downregulated VEGFA and HIF1α	[52], 2018
Cervical cancer	800 mg of EGCG per day for 4 months	98 women with HPV infection and CIN1	Did not promote the clearance of persistent high-risk HPV and related CIN1	[155], 2014.
Ovarian cancer	500 mL of EGCG-enriched green tea drink per day until recurrence or during a follow-up of 18 months	10 women with FIGO stage III-IV serous or endometrioid ovarian cancer	No significant health effects and no toxicity	[156], 2014
	10, 30, or 50 mg/kg of EGCG per day for 3 weeks	35 BALB/c nude mice	Inhibited tumor growth, upregulated PTEN, and downregulated PDK1, p-AKT, and p-mTOR	[106], 2020
	12.4 g/L of green tea drink per day for 60 days	10 athymic nude mice	Inhibited tumor growth through a reduction in ETAR and ET-1 expression	[108], 2006

MAPK-1, mitogen-activated protein kinase 1; NMNAT, nicotinamide nucleotide adenylyltransferase; VEGF, vascular endothelial growth factor; MMP-9, matrix metalloproteinase 9; EGFC, endothelial growth factor C; LH, luteinizing hormone; FSH, follicle-stimulating hormone; DHEA-S, dehydroepiandrosterone sulfate; SHBG, sex hormone-binding globulin; CRP, C-Reactive Protein; NOD1, nucleotide-binding oligomerization domain-containing protein; NAIP, neuronal apoptosis inhibitory protein; HIF1α, hypoxia-inducible factor 1-alpha; CIN1, low-grade cervical intraepithelial neoplasia; HPV, human papillomavirus; PDK1, phosphoinositide-dependent protein kinase; ET-1, endothelin-1; ETAR, endothelin A-type receptor.

**Table 3 nutrients-16-00559-t003:** Summary of registered clinical trials on EGCG treatment for selected gynecological diseases found on clinicaltrails.gov (31 November 2023).

Trial Number; Location	Title of the Study	Number of Participants; Type of Study; (Status)	Intervention	Outcome Measures
Uterine fibroids				
NCT05448365 Italy	The Use of Vitamin D in Combination With Epigallocatechin Gallate, D-chiro-inositol and Vitamin B6 in the Treatment of Women With Uterine Fibroid	60; Randomized, phase III trial; (recruiting)	EGCG, 300 mg; D-chiro-inositol, 50 mg; vitamin B6, 10 mg; and vitamin D, 50 μg, per day for 3 months	the total volume of UFs, the expression of ki67, ER, and PR, and the phosphorylation of the VEGF receptor
NCT05409872 Italy	Effects of Vitamin D, Epigallocatechin Gallate, Vitamin B6, and D-chiro-inositol Combination on UFs: a Randomized Controlled Trial	108; randomized; (recruiting)	EGCG, 300 mg; D-chiro-inositol, 50 mg; vitamin B6, 10 mg; and vitamin D, 50 μg, per day for 3 months	fibroid symptoms, quality of life, and volume of the fibroids
NCT05364008 United States	UFs and Unexplained Infertility Treatment With Epigallocatechin Gallate; A Natural Compound in Green Tea (FRIEND)	200; randomized; phase III; (recruiting)	Green tea extract, 1650 mg/day (45% EGCG), for up to 7 months with a 3-month run in the period, followed by ovarian stimulation with clomiphene citrate	the cumulative live birth rate, the conception rate, the miscarriage rate, and the fibroid volume
NCT04177693 United States	Pharmacokinetics and Hepatic Safety of EGCG	36; randomized; (recruiting completed)	EGCG, 800 mg alone, with 100 mg of clomiphene citrate, or with 5 mg of letrozole per day for 2 months	serum EGCG, EGC, 4′-O-methyl-epigallocatechin, total bilirubin, ALT/AST, alkaline phosphatase, βhCG, FSH, LH, AMH, estrogen, and folate levels and endometrial thickness
Endometriosis				
NCT02832271 China	Green Tea Extract for Endometriosis Treatment	185; randomized phase II, (completed)	EGCG, 800 mg per day for 3 months	changes in endometriotic lesion size, endometriotic growth, pain scores, quality of life, and total number and density of neovasculatures
Ovarian cancer				
NCT00721890, Canada	Green Tea Intake for the Maintenance of Complete Remission in Women With Advanced Ovarian Carcinoma (DBGT-OC-CR)	16; non-randomized; phase II trial (completed)	500 mL of EGCG-enriched green tea drink (double-brewed green tea) per day until recurrence or during a follow-up of 18months	no significant health effects and no toxicity
Cervical cancer/HPV infection/cervical lesions				
NCT05625308 Italy	Effect of Natural Compounds on the Severity of HPV-induced Cervical Lesions	40; non-randomized; interventional (completed)	EGCG, 200 mg; hyaluronic acid, 50 mg; vitamin B12, 1 mg; and folic acid, 400 mcg, per day for 12 weeks	regression of cervical lesions
NCT06098456 Italy	Epigallocatechin Gallate and Other Antural Compounds in HPV Infections	178; non-randomized, phase II, (active, not recruiting)	EGCG, 200 mg; hyaluronic acid, 50 mg; vitamin B12, 1 mg; and folic acid, 400 mcg, per day for 6 months	regression of cervical lesions, HPV infection, and lesion-related symptoms
NCT00303823 United States	Green Tea Extract in Preventing Cervical Cancer in Patients With Human Papillomavirus and Low-Grade Cervical Intraepithelial Neoplasia	98; randomized phase II trial; (completed)	800 mg of EGCG per day for 4 months	regression of cervical lesions

EGC, epigallocatechin; ALT, alanine aminotransferase; AST, aspartate aminotransferase; βhCG, human chorionic gonadotropin; FSH, follicle-stimulating hormone; LH, luteinizing hormone; AMH, antiMullerian hormone; HPV, human papillomavirus.

### 3.2. Endometriosis

Another benign gynecology condition is endometriosis, which is caused by the ectopic occurrence of glandular cells and endometrial stroma outside the uterine cavity [157]. These cells, similarly to the eutopic cells of the uterine mucosa, are able to respond to hormonal changes occurring during the menstrual cycle. Endometriosis implants are typically found within the minor pelvic cavity and peritoneum but may also occur in distant locations [157,158]. Endometriosis foci are usually divided into superficial implants, deep infiltrating endometriosis, and endometrial cysts [159]. The most common locations of lesions include the uterosacral ligament, ovary, ovarian fossa, vesicouterine pouch, and recto-uterine pouch. Endometriosis is a hormone-dependent disease with an inflammatory component, primarily affecting women of reproductive age. According to various authors, its frequency ranges from 5 to 15% in this age group [160]. The main symptoms associated with the presence of the disease include menstrual disorders, chronic pain, and infertility, which result in a significant reduction in the quality of life of the patients [161].

Various hypotheses have been formulated for the development of endometriosis, such as implantation, metaplastic, and other hypotheses, but the retrograde menstruation theory still has the most supporters [162,163]. It assumes that fragments of the uterine mucosa may travel backwards through the fallopian tubes to the peritoneal cavity, where they may be implanted, and then grow and infiltrate adjacent structures. Nevertheless, it is now known that this theory only partially explains the pathogenesis of the disease. According to the literature, the most frequently mentioned factors predisposing to the occurrence of endometriosis are reproductive age, early menarche, shortened cycles, a positive family history, being tall, or a diet rich in fats [163,164]. Endometrial tissue may appear in scars after surgery, including the removal of the appendix or inguinal hernia, amniocentesis, and laparoscopic procedures. The occurrence of endometriosis foci in the scar after a cesarean section is estimated at 0.03–0.4% of cases. The scar endometriosis can mimic carcinoma and presents a diagnostic challenge [165]. The formation of endometriosis after a cesarean section may occur through continuity or implantation. The mucosa of the uterine cavity, growing in the scar of the anterior uterine wall, may continuously invade all layers of the abdominal wall. Also, by moving fragments of the mucosa of the uterine cavity to the abdominal area with the operator’s hand or a surgical instrument, endometriosis may develop in the postoperative scar [166]. Due to its complex nature, endometriosis is difficult to diagnose. The average time from the onset of the first symptoms to the final diagnosis may even be about 7 years. Imaging tests used in the diagnosis of endometriosis primarily include transvaginal, transrectal, and transabdominal ultrasound, as well as magnetic resonance imaging. An ultrasound, especially transvaginal, is critical for identifying and evaluating possible lesions in the ovaries, including endometrial cysts. The absence of signs of the disease upon physical examination, ultrasound, and magnetic resonance imaging does not exclude the presence of the disease in a patient [161,167,168]. Only patients who have symptoms associated with endometriosis are eligible for treatment. An incidental diagnosis of endometriosis foci, e.g., in a laparoscopy performed for another reason, in a woman who does not report any problems typical of endometriosis, is not an indication for introducing therapy. Therefore, the treatment strategy depends on the severity of symptoms, age, reproductive plans, and the individual preferences of the patient. The main goal of treatment is to alleviate the symptoms, pain in particular, as well as preserve or restore fertility. Treatment may be divided into: conservative, surgical or combined. Pharmacological treatment is most often symptomatic and includes pain therapy and hormonal treatment [160,167].

#### 3.2.1. Epigenetic Mechanisms in Endometriosis Development—An Overview

The development of endometriosis may be associated with improper DNA methylation, histone methylation/acetylation, and the involvement of miRNA molecules. As a result of these epigenetic changes, modifications that occur in gene transcription co-regulators (activators, repressors, and enhancers) may result in the emergence of progesterone resistance at the stage of endometrial differentiation [169]. It has been proposed that the occurrence of additional epigenetic changes is influenced by microenvironmental signals such as hypoxia, pro-inflammatory cytokines, and locally produced estradiol. Over 40,000 CpG islands have been identified that differ in the level of methylation in endometriosis compared to healthy tissues [170,171]. It has been shown that the level of DNA methylation enzymes differs in the case of ectopic endometrium in patients with endometriosis compared to normal tissues. Higher expression levels of DNMT1, DNMT3A, and DNMT3B were found in ectopic endometriums compared to eutopic endometriums in women with endometriosis as compared to controls. Moreover, the expression levels of DNMT1, DNMT3A, and DNMT3B were shown to be positively correlated to each other [172]. The lower levels of acetylated H3 and H4 histones observed in cells with an endometriosis phenotype compared to normal endometrial stromal cells prove that aberrant histone modifications may cause the development of this disease. Reduced acetylation levels in the promoter region of genes responsible for cell cycle induction and proliferation were observed along with increased HDAC activity in endometrial cells [173,174]. Differences in the expression of HDAC1 and 2 were found in normal endometrium compared to endometriotic cells [175]. Moreover, it has been proven that the treatment of endometriotic cells with HDAC inhibitors causes the acetylation of histones H3 and H4 in the promoter region of cyclin-dependent kinase (CDK) genes, which leads to the blocking of cell proliferation [174,176]. For instance, apicidin, a novel histone deacetylase inhibitor, has profound antigrowth activity in human endometrial and ovarian cancer cells [170,176]. At the level of miRNA molecules, it has been shown that miR-135 participates in the regulation of, among others, HOXA10, a gene that is suppressed in endometriosis. The involvement of miR-199 has also been proven, which controls many inflammatory pathways, including IL-8, whose concentration is increased in patients with endometriosis [177].

#### 3.2.2. EGCG Treatment of Endometriosis

The beneficial effects of EGCG on the endometrium are mainly provided by its anti-angiogenic, antifibrotic, antiproliferative, and pro-apoptotic actions. Based on the results from mouse models, EGCG may inhibit the growth of endometrial lesions and affect the expression of E-cadherin [147,148]. EGCG significantly inhibits the development of experimental endometriosis through anti-angiogenic effects, which are observed as lower microvessel size and density and the downregulation of mRNA for angiogenic VEGFA [147]. EGCG treatment inhibits the development and reduces the size of endometriotic lesions by reducing cell proliferation and increasing apoptotic activity [100]. Moreover, even better bioavailability and higher antioxidation and anti-angiogenesis capacities were found for pro-EGCG compared to EGCG [148]. In vitro studies showed that EGCG suppresses E(2)-stimulated activation, proliferation, and VEGF expression in endometrial cells [101,154]. EGCG significantly inhibited the cell proliferation, migration, and invasion of endometrial and endometriotic stromal cells in patients with endometriosis. This treatment resulted in an inhibited TGF-β1-stimulated activation of the MAPK and Smad signaling pathways in endometrial and endometriotic stromal cells [99]. Methylation in the promoter region of E-cadherin was associated with endometriosis [178]. A group of mice administrated with EGCG had higher mRNA expression and a higher rate of promoter methylation of the E-cadherin gene than the control group [146]. As the downregulation of E-cadherin expression was related to Wnt/β-catenin pathway activation in endometrial cells, it may be proposed that EGCG action through epigenetic mechanisms results in the upregulation of E-cadherin in an endometrial lesion [179]. At this time, no clinical trials have confirmed this mechanism of action or shown the effects of EGCG on endometriosis.

### 3.3. PCOS

Polycystic ovary syndrome (PCOS) is a clinically heterogeneous disease whose incidence ranges, according to various sources, from 4 to 20% [180]. Based on the modified Rotterdam criteria, PCOS is diagnosed if any two of the following are present: (1) clinical and/or biochemical symptoms of hyperandrogenization, (2) absence or rare occurrence of ovulation, and (3) polycystic-appearing ovarian morphology on ultrasound, with the exclusion of other relevant disorders [181]. In addition to hormonal and environmental factors, there are many suggestions for an important genetic component in the etiopathogenesis of this disease [182]. However, to date, it has not been possible to clearly determine the gene or genes whose mutations or other structural changes could be responsible for the occurrence of the disease [183]. Genetic research on the pathogenesis of PCOS focuses on the pathways responsible for the secretion and action of insulin, mainly the insulin receptor (IR) gene, insulin (INS), and the insulin-like growth factor IGF and its receptor. Moreover, genes determining the activity of cytochrome P450 (CYP17, CYP11α) and genes of the androgen receptor (AR), LH receptor, leptin, follistatin, and many others are also taken into account [184]. Polycystic ovary syndrome lacks one dominant symptom. Due to differences in the ages of patients, other disease symptoms may arise. These include menstrual disorders, acne, infertility, hirsutism, or male pattern baldness [185]. The complexity of symptoms and individual differences in the clinical picture indicate the involvement of many metabolic pathways, resulting not only from the participation of genes that determine them, but also from complex transformations of the products of these genes. Polycystic ovary syndrome is the most common cause (70%) of infertility related to anovulation or abnormal menstruation. Mechanisms that may contribute to the development of PCOS include abnormal ovarian steroidogenesis, abnormal ovarian stimulation, internal defects of ovarian tissue, adrenal and ovarian hyperandrogenism, insulin resistance, and abnormal insulin secretion [186]. A combined contraceptive pill containing gestagen is usually prescribed to increase the frequency of periods and the number of ovulations and to treat hirsutism. Due to the fact that obesity is the most important environmental risk factor for PCOS, the treatment is aimed at lifestyle modification, as lowering body weight reduces insulin resistance. Additionally, specific diets, such as a low-GI diet or a ketogenic diet, may provide beneficial effects for these patients. Balanced sources of vitamins and microelements in the diet also support proper insulin management [187].

#### 3.3.1. Epigenetic Mechanisms in PCOS Development—An Overview

Abnormal histone modifications, DNA methylation, or transcriptional regulation by non-coding RNAs may contribute to the development of PCOS or PCOS-like phenotypic changes, including reproductive and metabolic abnormalities [188]. In women suffering from PCOS, epigenetic changes occur in ovarian tissue, peripheral and umbilical blood, and adipose tissue [189,190,191]. Abnormal DNA methylation may impact the expression of genes involved in cellular processes, such as lipid and steroid synthesis and sugar metabolism, which contribute to the pathogenesis of PCOS [191,192,193]. Because ovarian tissue from women with PCOS is difficult to obtain, there are only a few published works describing DNA methylation profiles in this tissue. In one study, hypermethylated genomic regions in women with PCOS were preferentially distributed at the edges of CpG and in promoters with a high CpG content, while hypomethylated genomic regions were found in gene bodies [194]. Different results were presented in a second study, where CpG islands and CpG island margins were hypomethylated in women with PCOS [195]. However, no significant relationship between DNA global methylation and PCOS incidence was reported [196].

Increased DNA methylation in the CYP19A1 promoter (encoding aromatase, the rate-limiting enzyme of estrogen biosynthesis) has been described in the context of the impaired androgen metabolism observed in women with PCOS. To date, however, it has not been clearly confirmed that changes in DNA methylation are the main reason for the differential gene expression observed in the ovarian tissue of women with PCOS [194,195].

Published data indicate that DNA methylation is increased in the promoter region of peroxisome proliferator-activated receptor gamma 1 (PPARGC1A), which suppresses its expression. Reduced PPARGC1A expression was also found to be associated with insulin resistance and high serum androgen levels in women with PCOS. Moreover, a lower level of DNA methylation of the LHCGR gene promoter has been observed in PCOS, which leads to an overexpression of the gene, resulting in increased LH levels in GCs, associated with the occurrence of gonadotropin disorders/modulation in women with PCOS. Conversely, the DNA methylation level of the LHCGR gene promoter is reduced in PCOS, and its overexpression leads to increased LH in GCs, which in turn leads to gonadotropin disorder in PCOS women [197]. Detailed assessments of DNA methylation at genomic loci in granulosa cells have shown hypomethylation in the LHCGR gene (encoding the LH receptor), NCOR1, and HDAC3 promoters, which correlates with LH receptor overexpression and dysregulated hormonal signaling [198,199]. In addition, the hypomethylation of CpG sites in untranslated LINE1 (also known as long-interspersed nucleotide transposable element 1) has also been reported, which may correspond to general DNA hypomethylation in granulosa cells [200]. Excessive methylation was observed in the promoter area of the CYP19A1 and PPARG genes, which may be related to the inhibition of the expression of specific genes involved in androgen metabolism [198].

A recently published study showed that the hypomethylation profile in PCOS patients was significantly different than that of the control subjects [201]. Modifications in histone H3 (acetylation) and in H3K9 (methylation) result in a reduction in the expression of CYP19A1, lowering cytochrome P450 aromatase activity, which could contribute to the hyperandrogenic phenotype and subsequent pathology in PCOS [201]. Data obtained in animal studies recognized that treatment with green tea extract has a lowering effect on the serum levels of LH, β-estradiol, and testosterone [202]. EGCG significantly inhibited the proliferation, steroidogenesis, and VEGF production of swine granulosa cells [203]. Beneficial effects on insulin resistance, blood glucose, and ovarian function were also found [204].

#### 3.3.2. EGCG in the Treatment of PCOS

The consumption of green tea by overweight and obese women suffering from PCOS leads to weight loss, a decrease in fasting insulin, and a decrease in the level of free testosterone [151]. Other researchers confirmed that green tea extract has beneficial effects on BMI, body weight, waist circumference, and body fat percentage, as presented in Table 2 [144,145,146,147,148,149,150]. On the contrary, a study on Chinese obese women with PCOS indicated that green tea as a complementary medicine does not have a significant effect on reducing weight loss [152]. A recently published review suggested that green tea may be a potential therapeutic option to attenuate PCOS complications, mainly due to its effect on weight loss and glycemic levels [205].

## 4. EGCG Treatment in Malignant Gynecological Diseases

### 4.1. Cervical Cancer

Cervical cancer is the fourth deadliest cancer among women worldwide, with one of the main risk factors being exposure to the human papillomavirus (HPV) [206]. E6 and E7, specific HPV oncoproteins, inactivate tumor suppressor proteins, such as p53 and retinoblastoma, and induce viral DNA replication, causing cancer proliferation [207]. Proteins involved in the EMT, such as E-cadherin and vimentin, were recognized in cervical cancer progression and metastasis [208,209]. Moreover, the EMT process involves the downstream phosphorylation of the Smad2 and Smad3 transcription factors determined by binding TGF-β to TGF-β receptors [210].

#### 4.1.1. Epigenetic Mechanisms in Cervical Cancer Development—An Overview

Moreover, women with folic acid deficiency, which is necessary for DNA synthesis, experience dysplastic changes in the cervix. These changes disappeared with folate supplementation, suggesting that folate deficiency may play a role in the development of cervical cancer, possibly through defective DNA synthesis and facilitating the persistence of the HPV virus [211,212]. The existence of an epigenetic mechanism of action has not been demonstrated here, but it is likely due to the participation of folic acid in providing a methyl group donor for the DNA methylation reaction.

#### 4.1.2. EGCG Treatment of Cervical Cancer

Towards cervical cancer, EGCG action mainly resulted in inhibited cell proliferation and induced apoptosis, as presented in Table 1 and Table 2. Green tea extract was reported to suppress TGF-β-induced epithelial–mesenchymal transitions via reactive oxygen species (ROS) generation, as well as inhibit proliferation and induce apoptosis via the VEGF, NF-KB, and Akt pathways in cell lines or mice with HeLa xenografts [110,213]. Recently published results showed that EGCG reversed the effect of TGF-β in cervical cancer cell lines by decreasing Smad2 and Smad3 phosphorylation and Smad activity and reducing reactive oxygen species (ROS) levels [110]. ROS generation, inhibited proliferation, and induced apoptosis via the VEGF, NFKB, and Akt pathways are mechanisms enhanced by green tea extract which aim at abolishing TGF-β-induced epithelial–mesenchymal transitions in mice xenografts with HeLa cells [110,213]. EGCG reduced DNMT to regulate DNA hypomethylation, induced cell cycle arrest, and downregulated p53 expression to regulate proliferation and apoptosis [118,120,214]. EGCG inhibited hypoxia-induced HIF-1α to inhibit VEGF, thus regulating angiogenesis [126]. Apoptosis induction and telomerase inhibition are other confirmed, beneficial actions of EGCG in preventing cervical cancer cell growth [215,216]. Cisplatin-based chemotherapy with concurrent radiation therapy is a commonly used strategy in cervical carcinoma, but the result is unsatisfactory due to its chemoresistance. An EGCG and cisplatin co-treatment presents synergistic anticancer activity [217]. EGCG could enhance the efficacy of cisplatin chemotherapy, resulting in inhibiting the proliferation of HeLa cells rather than cisplatin alone [218]. By evaluating the effect of EGCG on enzymes involved in epigenetic processes, it was shown that the transcriptional activity of DNMT and HDAC was significantly reduced in HeLa cells treated with EGCG. The exposure of HeLa cells to EGCG through changes in the methylation of promoter regions resulted in the reactivation of tumor suppressor genes, including RARβ, cadherin 1 (CDH1), and DAPK1. The RARβ, CDH1, and DAPK1 genes were found to be hypermethylated in untreated HeLa cells, and EGCG treatment reversed the promoter methylation of those genes [118]. HeLa cells treated with EGCG and 5-Aza-dC showed a decreased activity of DNMTs in a time-dependent manner. Moreover, a significant reduction in the expression of DNMT3b but no visible change in HDAC1 expression was reported. The results from a theoretical molecular model confirm that EGCG interacts with DNMT3B and HDAC1 [118]. Moreover, regarding epigenetic mechanisms, EGCG suppressed cervical carcinoma cell growth via regulating the expression of miRNAs. In addition to the inhibition of HeLa cell growth by EGCG in a dose- and time-dependent manner, it was significantly downregulated, and miR-210 and miR-29 were significantly upregulated [113]. In a review by Srivastava et al., the aberrant expression of miRNA was summarized in gynecological cancers [219]. The results indicate that miR-26a, miR-34c, miR-203, and miR-143 are downregulated, while miR-10a, miR-135b, miR-222, and miR-494 are upregulated in ovarian cancer. However, a panel of specific miRNAs with abnormal expressions confirmed for cervical cancer has not been proposed to date.

### 4.2. Endometrial Cancer

Endometrial cancer (EC) is another type of common cancer of the female genitals. Key risk factors include obesity, estrogen exposure, age above 55 years, early menarche, and late menopause [220]. The first subtype of EC is associated with excessive growth of the endometrium and is determined by estrogen stimulation and obesity. Alterations in the tumor suppressor–phosphatase and tensin homolog (PTEN), proto-oncogene Kirsten rat sarcoma virus (K-RAS), and β-catenin genes and microsatellite instability are involved in its molecular pathogenesis. The second type usually includes estrogen-independent serous carcinomas, which develop in the atrophic endometrium and, due to their lower differentiation, have a worse prognosis [221]. In type II, nuclear atypia and dysfunction of the p53 gene are frequently observed [222].

#### 4.2.1. Epigenetic Mechanisms in Endometrial Cancer Development—An Overview

Possible epigenetic mechanisms of endometrial cancer development and potential therapeutic targets have been recently reviewed [223,224]. The decreased promoter methylation frequency of VEGF, TNF, and HIF1α genes related to increased gene expression was noticed in endometrial cancer tissues as compared to a control study group [224,225]. Furthermore, genes frequently silenced by DNA promoter methylation in endometrial carcinoma were involved in DNA repair: MGMT and hMLH1 [226]. Aberrant methylation profiles of the tumor suppressor genes p16, PTEN, and RASSF1A and the steroid receptor genes Erα and PR-B were observed in endometrial tumorigenesis [227]. A direct association between PR gene expression and gene methylation was confirmed in endometrial cancer cells. The effect of promoter hypermethylation of PR-B and decreased PR expression was reversed by the DNMT inhibitor 5-aza-cd as well as the HDAC inhibitor trichostatin A [228]. Psilopatis et al. described the results of over 30 studies, which prove that HDAC inhibitors may become promising therapeutic agents for EC. These compounds inhibited tumor growth by inducing the transcription of silenced genes and by causing cell cycle arrest and apoptosis. In addition, in EC cells, a beneficial synergistic cytotoxic effect of HDAC with traditional chemotherapeutics was found [229].

The occurrence of EC is associated with the deregulation of a number of miRNA molecules [230]. It seems that the miRNA panel that may have the greatest diagnostic value for EC includes the following molecules: miR-205, miR-200 family, miR-135b, -182, -183, and -223. miR-205 was most consistently found to be upregulated and affect the expression of the tumor suppression gene PTEN, which leads to a reduction in cell apoptosis and the development of EC. In addition, miR-205 inhibited the tumor suppressor gene JPH4 (Junctophilin 4), contributing to the metastasis and tumor progression of EC [231]. In addition, associated with malignant cell transformation, miR-200 family members like miR-200a, miR-200b, and MiR-200c have been reported to be upregulated in EC [232]. Another confirmed association of miR molecules with EC relates to the upregulation of miR-223 [233]. miR-223 targets insulin growth factor receptor 1 (IGF-1R), whose reduced expression has been observed with aggressive EC and a worse prognosis for this type of tumor. miR-182, which induces cell proliferation by targeting tumor suppressor gene transcription elongation factor A-like 7 (TCEAL7), was also found to be upregulated in EC [234]. In endometrioid endometrial carcinoma, the upregulation of miR-183-5p was related to the downregulation of the cytoplasmic polyadenylation element-binding protein 1 gene (CEBP1). Among the miR-183 targets, CEBP1 functions as a tumor suppressor, MMP-9 is involved in ECM remodeling, and FOXO1 regulates the cell cycle [230,235].

#### 4.2.2. EGCG Treatment of Endometrial Cancer

Regarding endometrial cancer, the effect of the EGCC prodrug (pro-EGCG) exhibited antiproliferative, pro-apoptotic, and antitumor actions via the ERK/JNK, Akt, and PI3K/AKT/mTOR/HIF1α signaling pathways [51]. Antiproliferative, pro-apoptotic, and anticancer effects were reported to occur via the ERK/JNK, Akt, and PI3K/AKT/mTOR/HIF1α signaling pathways in EGCG-treated endometrial cancer cells [52]. In vitro and in vivo studies have confirmed the beneficial effects of EGCG and proposed several pathways that this compound regulates; however, the detailed mechanism of action is still unknown. By downregulating vascular endothelial growth factor A and HIF-1α, pro-EGCG participated in the inhibition of tumor angiogenesis in xenograft animal models of endometrial cancer [52]. The effect of EGCG on hormonal regulation was confirmed by the observation of a decrease in the expression of estrogen and progesterone receptors in endometrial adenocarcinoma cells [105]. A reduction in EC risk from green tea consumption has not been conclusively confirmed, and the exact molecular mechanisms have not been elucidated. In a recently published paper, tea had no protective effect against EC [236]. An earlier study also failed to prove that tea intake may have a beneficial therapeutic effect on ovarian cancers [237]. The inconsistent results from human and in vitro studies may arise from the poor bioavailability of EGCG from green tea in organisms, and therefore, a system for the efficient delivery of EGCG to cells is needed. Despite the number of studies on the role of miRs, there are still no evidence on the effect of EGCG on the regulation of miRs in endometrial cancer cells. A more thorough analysis is needed to determine how the different miR modulations by EGCG relate to the effects of epidemiological studies related to the consumption of foods containing these polyphenols.

### 4.3. Ovarian Cancer

Ovarian cancer is one of the leading causes of cancer deaths among women. With regard to the risk factors, age, familial cancer syndrome, and mutations in the breast cancer susceptibility gene (BRCA), as well as smoking, alcohol consumption, a lack of physical activity, obesity, and an unhealthy diet, are proposed. A total of 90% of diagnosed ovarian cancers are of epithelial origin, and the treatment is focused on surgery and platinum and taxane cytotoxic chemotherapy [237]. Ovarian cancer cells arise from epithelial, stromal, or germ cells in the ovary [238]. The cells disseminate through the perineal cavity and metastasize to the omentum. Ovarian cancer is a complex disease, and due to its non-specific clinical symptoms, it is usually detected in advanced stages. Among treatment options, combining surgery with chemotherapy is the gold standard, but growing evidence has shown the effectiveness of targeted therapies against ovarian cancer, including anti-angiogenesis therapies and immune checkpoint inhibitors [239].

#### 4.3.1. Epigenetic Mechanisms in Ovarian Cancer Development—An Overview

It was proposed that abnormal DNA methylation might be involved in the disruption of biological pathways essential for the transition of an ovarian tumor to a drug-resistant phenotype. The reduced susceptibility of ovarian cancer cells to cisplatin-mediated DNA damage is due to abnormal DNA methylation. Therefore, it was suggested that DNMT inhibitors might become part of a chemosensitization strategy [240]. A recently published paper confirmed that EMT genes involved in the metastasis mechanism were regulated by DNA methylation in cisplatin-resistant ovarian cancer cells and, therefore, could be a target of epigenetic-based therapy [241,242]. A significant relationship was reported between promoter methylation of the RAS association domain family protein 1a gene (RASSF1A), one of the tumor suppressor genes, and the risk of ovarian cancer [243]. Satellite 2 DNA sequences (Sat2) in the juxtacentromeric heterochromatin of chromosomes 1 and 16 are characteristic of long-interspersed nuclear elements (LINE1). These highly repetitive elements are hypermethylated in normal tissues and hypomethylated in cancers cells. Research into ovarian carcinomas has recognized characteristic hypomethylation in Sat2 [244,245]. Moreover, ovarian cancer cells were found to have a reduced LINE-1 methylation phenotype corresponding to the global methylation and hypermethylation of specific gene sites such as the BRCA1 promoter [246,247,248].

Aberrant DNA methylation and the overexpression of the BRCA2, NFKB1, CTGF, and BCL3 genes involved in cell proliferation have been found in ovarian cancer [249].

#### 4.3.2. EGCG Treatment of Ovarian Cancer

EGCG has an anticancer effect on ovarian carcinoma, mainly through the inhibition of different genetic signaling pathways which are closely linked with tumorigenesis [250]. Based on the results from xenograft models and ovarian cancer cells, the inhibition of tumor progression by EGCG was confirmed. This compound of green tea effectively acted on the nuclear factor kappa B subunit 1 (NFKB1), HIF1α, and MMP genes, which are involved in cell proliferation [251]. EGCG induced apoptosis in ovarian cells by activating DNA damage responses and oxidative stress [252,253,254]. Treatment with EGCG reduced cell proliferation and invasion [108]. As EGCG is recognized as a DNMT inhibitor, a decrease in DNA methylation could be one of its mechanisms of action. There is no confirmed evidence of the influence of lifestyle factors on the change in DNA methylation in ovarian cancer. However, an association was demonstrated between the methylation profile and exposure to sex hormones or obesity, which are known as the main risk factors for ovarian cancer [255,256].

The role of EGCG in overcoming drug resistance in ovarian cancer was confirmed and linked to specific signaling pathways. EGCG and sulforaphane administration downregulate the hTERT and BCL-2 genes, induce apoptosis, and promote DNA damage responses in paclitaxel-resistant ovarian cancer cells [257]. Additionally, a combination of those compounds enhanced the efficacy of cisplatin treatment by inducing apoptosis, arresting cells in the G2/M phase, and upregulating p21 expression in cisplatin-sensitive ovarian cancer cells [119].

Unfortunately, there are practically no studies analyzing the effect of EGCG administration on epigenetic mechanisms in ovarian, endometrial, or uterine fibroid cells. Therefore, the proposals for such an action are based on data from research on breast cancer cells, which, like a hormone-dependent tumor, are characterized by molecular factors determining their growth similar to those determining the growth of ovarian cancer or uterine fibroids [258]. Using an in vitro model of breast cancer, it was shown that EGCG administration reduced the EZH2-catalyzed trimethylation of histone H3 at lysine 27 (H3K27me3) with a corresponding increase in the deposition of transcriptionally active acetylated histone H3 at lysine 9/18 (H3K9/18 Ac), among other things observed in the TIMP-3 promoter region. This gene encodes a tissue inhibitor of metalloproteinase-3 (TIMP-3), which, by binding to proteinases, suppresses their activity and thus protects the extracellular matrix against degradation. TIMP-3 has anticancer properties, including the induction of apoptosis and antiproliferative, antiangiogenic, and antimetastatic effects [259]. EGCG administered to MDA-MB-231 and MCF-7 cell lines inhibited the migration and invasion of breast cancer cells. EGCG was shown to reactivate the expression of the SCUBE2 gene, a tumor suppressor gene that is often methylated and silenced in breast cancer. Additionally, lower expressions of the DNMT1, DNMT3a, and DNMT3b genes and DNA hypomethylation were found [260]. It has been shown that EGCG has the potential to reverse both DNA methylation and histone acetylation, inducing apoptosis in breast cancer cells and inhibiting cancer development. EGCG decreased the activity of DNMT1 and HDAC1 and promoted global histone H3 acetylation in breast cancer cells [261].

## 5. Summary

The polyphenols present in green tea, in addition to EGCG, include catechin, epicatechin, epicatechin gallate, and epigallocatechin. Due to their similar chemical structures, it is postulated that the mechanisms of their actions on cells are similar and include, among others, pro-apoptotic, antiproliferative, pro-oxidant, and antioxidant effects (Figure 2). Their antioxidant effects can be exerted directly though scavenging free radicals or metal chelation. Indirectly, catechins may affect pro-oxidant and antioxidant enzymes or signal transduction pathways involving TNF or NFKB [262]. Modifications of the secretion of inflammatory cytokines ultimately affect the process of angiogenesis and the proliferation of cancer cells. Since diseases of the reproductive system, like most diseases, are described as mediated by various inflammatory states, compounds that act by alleviating inflammation, such as those present in green tea, are proposed as health-benefit compounds/a current therapeutic option [263]. The proposed mechanisms by which EGCG and other catechins modulate cellular activity and the environment are complex. EGCG has been reported to interact with transmembrane receptors or proteins involved in disease progression, for example, TGF-β in cell differentiation and migration pathways or fibronectin in ECM formation. Many studies have proven the pro-apoptotic and antiproliferating effects of EGCG, but only a few have recognized their epigenetic mechanisms. Typically, the levels of methylation or expression of selected genes or the activity of selected enzymes catalyzing reactions important for epigenetic mechanisms are assessed. To answer the question of the actual impact of EGCG, the methylation patterns of all genes should be compared to indicate what changes have occurred precisely. Due to the relatively small number of studies exploring the epigenetic mechanisms of EGCG in gynecological cancers, we can rely on the published results regarding other types of cancer, where EGCG affects multiple molecular targets involved in cancer initiation, promotion, and progression. A good example that links epigenetic aberration to tumor development is reviewed by Dorna et al. [264]. The pro-apoptotic or antiproliferative effects of EGCG resulting from changes in gene expression observed in in vitro studies are most likely related to a modification of the action of methylation enzymes such as DNMTs and a modification of histone proteins. The most emphasized modification is the DNMT1 inhibition achieved by direct enzymatic inhibition, indirect enzymatic inhibition, or reduced DNMT1 expression and translation [118,265]. Knowing the effect of excessive gene methylation, we can predict the phenotypic effect in EGCG-treated cells.

In addition, the beneficial effect of EGCG on cells was further enhanced when vitamin D was taken at the same time [128]. Optimal levels of vitamin D have been shown to help prevent and regress low-grade squamous intraepithelial lesions of the cervix and reduce the risk of fibroids [266]. However, there is no clear statement of the exact mechanism. Proposed mechanisms of action include inhibiting cell proliferation, promoting apoptosis, and modulating inflammatory responses. Vitamin D3 increased apoptosis and necrosis and decreased the proliferation of eutopic endometrial stromal cells and ectopic endometrial stromal cells [267]. Because the vitamin D receptor functions as a transcription factor in reproductive cells and interacts with many transcription modulators, the modulation of their action by vitamin D may alter gene expression [268]. Another mechanism proposed to explain the action of VDR is the methylation of its gene [269]. A recently published study indicated an inverse correlation between the expression and hypermethylation of the VDR gene in PCOS patients [270]. Vitamin D3 treatment induced H3K4me3 protein expression and exhibited antiproliferative effects in ovarian cancer cells, which could explain the effect of calcitriol on tumor suppression [271]. Experiments on a cell culture trophoblast model showed a downregulation of H3K9ac after calcitriol treatment [272]. These studies prove that natural compounds and those derived from food can act through epigenetic mechanisms. Because green tea catechins are compounds with an acceptable safety profile, it seems that they could be included in dietary recommendations for many diseases, including those affecting the reproductive system.

## 6. Conclusions and Future Directions

Being a natural source from tea plants is an important advantage of EGCG. Therefore, consuming EGCG supplements or drinking green tea may be a safe dietary recommendation for UF patients. However, due to their poor bioavailability and rapid elimination, new EGCG derivatives should be sought, and the development of nanotechnology should be used for their more effective delivery to cells. Pre-clinical and clinical evidence clearly shows that EGCG has broad antioxidant, anti-inflammatory, and anticancer effects. Further studies, both in vivo and in vitro, using high-throughput methods are necessary to evaluate the molecular mechanisms responsible for the additive or synergistic effect of EGCG with other supplements or drugs used for UFs. Perhaps it would be worth researching other natural compounds and looking for those with which EGCG will present with the best effect—as an option for those patients who do not want to use hormonal therapy. The synergies between EGCG and hormonal therapy are also of great interest and await future trials in this matter. Epigenetic mechanisms, especially interactions between DNA methylation and gene expression, are studied intensively in the context of malignant but also benign tumors. Incorporating EGCG-based therapy into the treatment of symptomatic women with UFs may have long-term benefits. Further research is needed to assess the health effects of such therapy in women, as well as to understand the molecular mechanisms responsible for changing the methylation profile and gene transcription.

## Figures and Tables

**Figure 1 nutrients-16-00559-f001:**
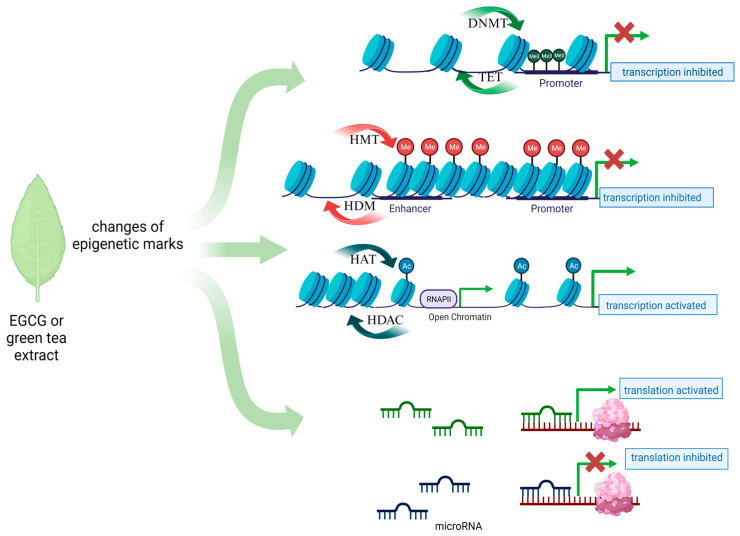
Main epigenetic mechanisms proposed for EGCG action in gynecological diseases.

**Figure 2 nutrients-16-00559-f002:**
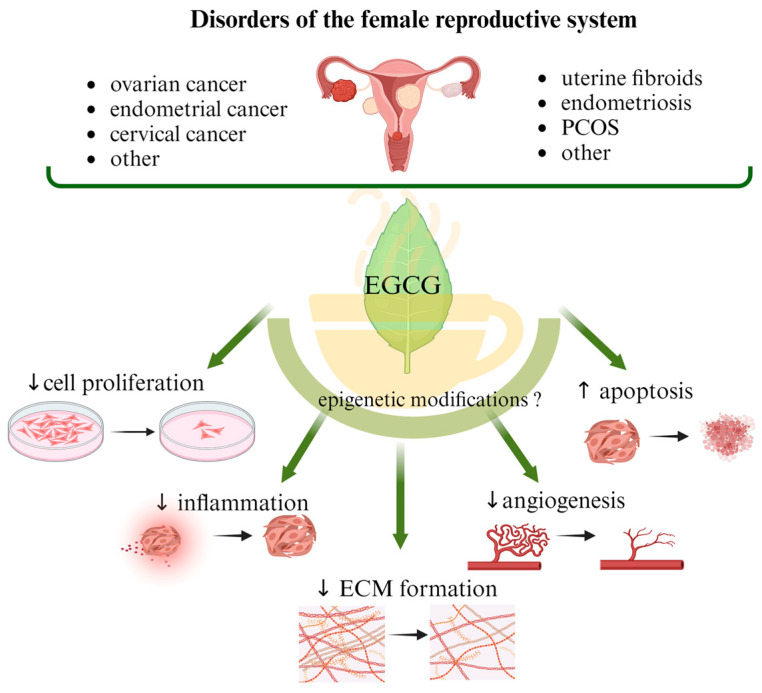
A summary of the proposed mechanisms of action of EGCG regarding female reproductive system disorders. PCOS, polycystic ovary syndrome; ECM, extracellular matrix.

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
