# Peer review of "Epigallocatechin Gallate for the Treatment of Benign and Malignant Gynecological Diseases—Focus on Epigenetic Mechanisms"

_nutrients, 2024, doi:10.3390/nu16040559_

Round 1

Reviewer 1 Report

Comments and Suggestions for Authors

Epigenetics is a science that studies changes in gene expression that are not related to changes in the sequence of nucleotides in DNA. Changes in gene activity can be modified by the influence of environmental factors and be inherited. Epigenetic mechanisms are based on the expression and suppression of genes encoding biologically active proteins that control metabolic processes, leading to the appearance of desired traits and functions. The most relevant mechanism of epigenesis is DNA methylation.

 In the study, the researchers focused on the role of epigenetic processes in the onset of the most common gynecological conditions, excluding pregnancy. There are data supporting the epigenetic effects of adverse factors acting in pregnant women on the health of their offspring.

 In the paragraph on endometriosis, I would suggest mentioning endometriosis in the scar of the anterior abdominal wall after a cesarean section. Abdominal shell endometriosis is an iatrogenic disease and is associated with intraoperative implantation of endometrial elements into the wound of the shells. The formation of tumors after cesarean section can be accomplished by continuity or implantation. There is also, although extremely rarely, a risk of neoplastic transformation of a focus of endometriosis in a cesarean section scar.

 Line 766 should properly cite the cited literature (Bleotu, C., et al, Interplay of Epigenetics with Gynecological Cancer, Gynecologic Cancers - Basic Sciences, Clinical and Therapeutic Perspectives, 2016.p...).

Author Response

  • Epigenetics is a science that studies changes in gene expression that are not related to changes in the sequence of nucleotides in DNA. Changes in gene activity can be modified by the influence of environmental factors and be inherited. Epigenetic mechanisms are based on the expression and suppression of genes encoding biologically active proteins that control metabolic processes, leading to the appearance of desired traits and functions. The most relevant mechanism of epigenesis is DNA methylation. In the study, the researchers focused on the role of epigenetic processes in the onset of the most common gynecological conditions, excluding pregnancy. There are data supporting the epigenetic effects of adverse factors acting in pregnant women on the health of their offspring.

Our Response:

We would like to thank you for these positive comments.

  • In the paragraph on endometriosis, I would suggest mentioning endometriosis in the scar of the anterior abdominal wall after a cesarean section. Abdominal shell endometriosis is an iatrogenic disease and is associated with intraoperative implantation of endometrial elements into the wound of the shells. The formation of tumors after cesarean section can be accomplished by continuity or implantation. There is also, although extremely rarely, a risk of neoplastic transformation of a focus of endometriosis in a cesarean section scar.

Our Response:

We would like to thank you for this important comment. We added information about the development of endometriosis in the scar after a cesarean section.

  • Line 766 should properly cite the cited literature (Bleotu, C., et al, Interplay of Epigenetics with Gynecological Cancer, Gynecologic Cancers - Basic Sciences, Clinical and Therapeutic Perspectives, 2016.p...)

Our Response:

We would like to thank you for this important comment. The mentioned citation has been corrected.

Reviewer 2 Report

Comments and Suggestions for Authors This is a good review for potential health benefits of epigallocatechin-gallate (EGCG) which is contained in green tea, in various kinds of gynecologic diseases focusing on mostly antioxidant action and epigenetic regulation.
The authors have described pathogenesis and treatment of each gynecologic diseases, and clearly reviewed the role of EGCG based on many reliable references. The manuscript is well-written and easy to read. The strength point of this review is proper length of summaries in each category. Thanks to these summaries, readers can gain not only basic knowledge of Epigenetics and EGCG, but also broad knowledge about various gynecological diseases from this single review.

However, provided figure is only one which explains a main epigenetic mechanism. The additional figures explaining for some gynecologic diseases with possible role of EGCG would improve quality of the paper, if possible.

Author Response

Reviewer 2

  • This is a good review for potential health benefits of epigallocatechin-gallate (EGCG) which is contained in green tea, in various kinds of gynecologic diseases focusing on mostly antioxidant action and epigenetic regulation. The authors have described pathogenesis and treatment of each gynecologic diseases, and clearly reviewed the role of EGCG based on many reliable references. The manuscript is well-written and easy to read. The strength point of this review is proper length of summaries in each category. Thanks to these summaries, readers can gain not only basic knowledge of Epigenetics and EGCG, but also broad knowledge about various gynecological diseases from this single review.

Our Response:

We would like to thank you for these positive comments.

  • provided figure is only one which explains a main epigenetic mechanism. The additional figures explaining for some gynecologic diseases with possible role of EGCG would improve quality of the paper, if possible.

 Our Response:

We would like to thank you for these important comments. Following the reviewer's suggestion, we have prepared a graphical abstract and figure summarizing the effect of EGCG on cell function discussed in the manuscript.

Reviewer 3 Report

Comments and Suggestions for Authors

This is an extensive review on EGCg and epigenic aspects on gynecologic diseases. However, this review needs to be conscise and focused on EGCg effects on epigenic modifications on gynecologic diseases. Readers might get helful information from the organized themes on EGCg or in comination with other supplements for their beneficial effects on combatting the gynecologic diseases. The following specific comments are:

1. The faint line between benign and maligant diseases of gynecological disorders are presented in this review article. It is not clear the importance of epigenic alterations in UFs and ovarian cancers, respectively. In addition, EGCg or other catechins or just drinking tea would provide the broad spectum for benefits on preventing benign and maligant gynecologic diseases,

2. The important information regarding EGCg acting to combat the gynecologic diseases have been provided in this reviwew article. However, how EGCg or green tea with and without other compounds act on histone acetylation or DNA methylation or gene expressions in diseased cells are not clear. It should be able to improve the clearness if the focus on the EGCg or green tea benefits on epigenetic alterations for reseucing apoptosis or inflammation in cells.

3. Are EGCg or other catechins or Vitamin D act on the same mechanism on curing UFs or other maligant gynecologic diseases by epigenetic modifications?

4. some typos are found in the manuscripts. 

5. Clinical trials numbers need to be specified and referenced. 

Author Response

Reviewer 3

This is an extensive review on EGCG and epigenic aspects on gynecologic diseases. However, this review needs to be concise and focused on EGCG effects on epigenic modifications on gynecologic diseases. Readers might get helpful information from the organized themes on EGCG or in combination with other supplements for their beneficial effects on combatting the gynecologic diseases. The following specific comments are:

  1. The faint line between benign and malignant diseases of gynecological disorders are presented in this review article. It is not clear the importance of epigenic alterations in UFs and ovarian cancers, respectively. In addition, EGCG or other catechins or just drinking tea would provide the broad spectum for benefits on preventing benign and malignant gynecologic diseases,
  2. The important information regarding EGCg acting to combat the gynecologic diseases have been provided in this review article. However, how EGCg or green tea with and without other compounds act on histone acetylation or DNA methylation or gene expressions in diseased cells are not clear. It should be able to improve the clearness if the focus on the EGCg or green tea benefits on epigenetic alterations for reseucing apoptosis or inflammation in cells.

Our Response:

We would like to thank you for your valuable comments. A number of studies have been published on the beneficial effect of EGCG on cell functions, especially in the prevention of cancer transformation. However, there are currently no studies globally assessing this effect in the body. Typically, the levels of methylation or expression of selected genes or the activity of selected enzymes catalyzing reactions important for epigenetic mechanisms are assessed. To answer the question about the actual impact of EGCG, methylation patterns of all genes should be compared to indicate what changes have occurred precisely. Due to the relatively small number of studies exploring the epigenetic mechanisms of EGCG in gynecological cancers, we can rely on the published results regarding other types of cancer, where EGCG affects multiple molecular targets involved in cancer initiation, promotion, and progression. Many studies have been published on the beneficial effect of EGCG on cell functions, especially in the prevention of cancer transformation. Currently, however, no studies globally assess this effect in the body, i.e., the results of the proteome, transcriptome, and methylome studies.

Ad2. We have added text fragments discussing this issue, emphasizing the important proven effects of EGCG or green tea.

Ad3. We also mentioned the mechanism of action of other catechins and vitamin D in relation to the discussed gynecological disease in the new paragraph at the end of the manuscript.

Ad4. We checked the text to correct any typos.

Ad5. We added a table with characteristics of registered clinical trials.   We hope this has increased the value of the manuscript.

Round 2

Reviewer 3 Report

Comments and Suggestions for Authors

The revised manuscript has been very much improved.